# Analysis of the Contribution of Land Registration to Sustainable Land Management in East Gojjam Zone, Ethiopia

Abebaw Andarge Gedefaw

Institute of Land Administration, Debre Markos University (DMU), Debre Markos 269, Ethiopia; abebaw.andarge@dmu.edu.et

**Abstract:** Land registration programs on a large scale aimed at strengthening the land rights of farm households in Ethiopia have been executed in different degrees across different regions since 1998. This study investigates the contribution of land registration on the perceived tenure security of farmers, farmer confidence, women and marginalized groups, and sustainable land-management practice after receiving a land holding certificate in the dryland areas of East Gojjam Zone, Ethiopia. Face-to-face interviews were conducted with 385 households selected by using stratified random sampling techniques. Furthermore, focus group discussions and key informants are primary data sources. According to an investigation of qualitative and quantitative data, 163 households have a mean of 0.40 ha of agricultural land on steep slope areas, and approximately 26% of households are afraid of land redistribution and farm loss in the next five years. Moreover, 22% of households fear the government taking their farm plot at any time. Respondents, on the other hand, believe that land registration has reduced the landlessness of women, the disabled, and the poorest of the poor while increasing the landlessness of youths. After land registration, household participation in land-management practices increased by 15%. Despite this, the difference in the mean of major crop yields per household is insignificant, except for wheat, which decreased significantly at the $p < 0.1$ level. The study determined household head age, household size, land management training and advice, livestock holdings, and the mean distance from farm to settlement as influential factors for increasing construction of water-harvesting systems. Land registration, in general, enhances land tenure security, land-management practice, and land rights of women and marginalized groups of societies, but did not improve crop productivity. The findings should persuade policymakers to address potential sources of insecurity, such as future land redistribution issues.

**Keywords:** land registration; sustainable land management; land tenure security; water-harvesting system; dryland areas; East Gojjam Zone; Ethiopia

## 1. Introduction

### 1.1. Background of the Study

Ethiopia is currently in the process of economic transformation with the goal of becoming a lower-middle income economy by 2025. Agriculture is arguably the most important focus of this process, as developing the agricultural sector is one of the best ways to stimulate rapid, inclusive economic growth. However, this development would have proven impossible if not for land registration. The International Federation of Surveyors (FIG) defines land registration as the official recording of legally recognized interests in land [1]. Land registration is important in understanding the impact of human societies on natural systems, which also has psychological implications [2]. Land degradation entails soil erosion [3,4], desertification [5], pollution [6] and inappropriate land-management practices [7], among others. Land degradation is also caused by human intervention in natural ecosystems [8,9]. Environmental and socioeconomic issues such as high population pressure, land degradation, unsustainable farming practices, and land tenure insecurity impede Ethiopia's agricultural development [10–14]. Across Africa, land tenure insecurity

limits agricultural production and livelihood improvements [15,16]. Government efforts to achieve their development goals can be hampered by tenure insecurity, which is seen to affect agricultural productivity [17–19].

Land is an essential component of household socioeconomic capital, especially in Africa, where agriculture supports most households. More importantly, secure access to land is critical. Long-term investments in sustainable livelihoods by rural households are required for sustainable agricultural development [20]. Most African communities rely on land for survival, and land resources are the cornerstone of achieving many of the UN Sustainable Development Goals (SDGs) [21]. Moreover, securing land rights has been identified as an important strategy for achieving SDGs [22]. The 2030 United Nations Development Goals, specifically Goal 1 (poverty), Goal 2 (hunger, food security, nutrition, and sustainable agriculture), Goal 5 (addressing gender equality and the empowerment of women and girls), and Goal 15 (issue on life on land), emphasize the importance of access to and control over land, as well as sustainable land management and associated resources. As a result, a modern land administration system, including formal land registration, titling, and certification, has been viewed as a prerequisite for ensuring property rights and agricultural development [23–25]. Land tenure should be properly administered for positive societal changes by establishing formal land titling procedures [26]. It is argued that tenure security has a positive impact on land investment by improving holding rights and providing a sense of stability, which encourages farmers to make sustainable land investments and increase yield [27]. The need to divert private resources to protect property rights is decreased by improved tenure security [28]. The main finding of empirical research is that land tenure security improves land-related investment [29–32] by strengthening land claims and enhancing farmers' credit access [33,34] and agricultural productivity [35,36]. Titling, on the other hand, can enhance intensification and other unsustainable land practices by fueling land contestation, particularly in legally pluralistic contexts [37–40], and reinventing local common-pool resource problems that communities may or may not be willing to address [41,42].

Contrarily, tenure insecurity is a major barrier to the adoption of sustainable land management, contributing to increased environmental degradation across Sub-Saharan Africa, including Ethiopia [43–47]. It has long been recognized that unclear and insecure property rights can discourage farmers from making land-improvement investments due to the uncertainty and future expropriation risk by the government [48–50]. Furthermore, theoretical and empirical studies suggest that a lack of secure access to land is frequently seen as a significant factor in food insecurity, limited livelihood opportunities, and, consequently, poverty [20,51].

Thus, calls for land titling are widespread and have been going on for a long time in Africa, despite the fact that early land reform programs were frequently unsuccessful [52,53]. However, the growing need in Africa for the formalization of land rights and a well-regulated land management system is highlighted by the increasing pressure on farmland brought on by population growth and foreign investor demand for large-scale agricultural land [54].

Due to the importance of land as a source of livelihood and political power in Ethiopia, the land tenure system has been at the forefront of policy debates for generations [55,56]. In the decades prior to and during the imperial era, land was concentrated in the hands of absentee landlords, and arbitrary evictions posed a serious threat to tenant farmers [56,57]. After overthrowing the imperial regime of Haile Selassie through a military coup (1974), the socialist Derg regime implemented radical reforms that altered the agrarian structure and access to land, transferring land ownership to the state [56,58].

Following the fall of the Derg regime in 1991, the current government began to liberalize the economy. However, the reform package largely "overlooked" the land issue mainly land reform, and the legacy of the Derg continued to define key elements of current land policy [55,56]. Land rights are still held by the state. On the other hand, the current administration has made several changes. First, responsibility for land issues was devolved

to regions. Second, the frequency of land redistribution (where the aim was to redistribute land according to the needs (family size) of households and to provide land for young married couples, women, marginalized groups, and youth) has been reduced, but it is not entirely off the agenda. Third, while land rentals are officially permitted, some regions still impose restrictions on the terms of rental contracts. Overall, the state continues to be a source of tenure insecurity. The government remains critical of privatizing land holdings, retaining a discourse of social equity and protection of land concentration in the hands of the few. However, some have argued that the government uses land rather as a "carrot and stick" to achieve political goals [55,56].

In the past, Ethiopia experienced frequent land redistribution, which led to land fragmentation, underutilization of land, and tenure insecurity [29,59–63]. Furthermore, land redistribution was primarily carried out in the years immediately following the 1975 governmental change, but additional land redistributions have occurred since then (constitutionally this requires a significant majority to demand a land redistribution to take place) [64]. As a result of these legal changes, and significant land holding shifts, smallholders did not perceive that they had a high degree of land tenure security—the land redistribution after all was only usufruct rights, not ownership rights. This tenure system was largely continued with the entrance of the new government in 1991, which made only minor changes to the ability to rent land on a short-term basis. In 1997, the Amhara National Regional State made significant land redistributions. Following this, there was much debate in Ethiopia about the consequences of this redistribution. Farmers have been discouraged from making improvements to their land due to the perception that land redistribution undermines tenure security [65]. Therefore, it is thought that this fragmentation and reallocation of land holdings will negatively affect land management activities [65]. Ethiopia's government is currently focusing on landscape restoration and sustainable land management.

In response to the negative effects of tenure insecurity on sustainable land-management practices, the Ethiopian government executed a large-scale land registration program in 1998. Ethiopia has one of Africa's most extensive, rapid, and low-cost land registration reforms, and has been cited as a model for land certification in Africa [66]. Across the four regions (i.e., Amhara, Oromia, Tigray and (SNNP) Southern Nations, Nationalities, and Peoples), some 15 million parcels of the total 50 million parcels had been registered and certificates distributed to landholders. From these, about 25% of the parcels are solely owned by women and 55% jointly held by husbands and wives. Only 20% of the total parcels registered were under name of male landholders alone [67]. Previous research on the effects of land registration in Ethiopia has focused primarily on the Tigray region. According to these studies, land registration is associated with higher levels of land-related investment and productivity [68], improved welfare [58], increased land rental market participation [58] and reduced border conflicts [69]. Similar to this, the Amhara region of Ethiopia has also documented the positive and significant effects of land registration on household perceived tenure security, investment, and land market participation [29]. In addition to this, [57,66] used data from four major regions of Ethiopia and discovered that land registration has a positive effect on land management.

According to one of the preambles of the Amhara Region Rural Land Administration and Land Use Proclamation No. 133/2006, the establishment of land ownership enhances landholders' ability to use their labor, wealth, and creativity [70]. Any person granted rural land in the region shall be given a land holding certificate, on which the details of the land are registered by the Authority and his photograph is fixed [70]. However, previous studies focused on the effect of land registration on tenure security and land-management practice by comparing titled and untitled land holders at the kebele level. Moreover, Gedefaw et al. [71] focused on the effects of land certification on sustainable land management, particularly on terracing and manure use. Another study carried out by Mengesha et al. [25] investigated land certification effect on sustainable land management, especially on tree planting. One key exception is [72], who studied the effect of land

certification on sustainable land-management practice in the dryland areas. Farming in dryland areas is risky due to lack of rainfall and unsustainable land-management practices. Still, no study on the contribution of land registration on sustainable land management has focused on the construction of water-harvesting systems in the dryland areas of Ethiopia generally, and in the Amhara region specifically. However, this study aimed to fill the existing research gap by investigating, with reasonable scientific justification, the changes brought about by individual households in terms of land tenure security and land management before and after registration in the dryland area of Ethiopia's East Gojjam Zone. Therefore, this study investigated the contribution of land registration on perceived tenure security of farmers, farmer confidence, women and marginalized groups, and sustainable land-management practice in dryland areas.

To achieve this objective, the following research questions were formulated:

a. Does land registration improve the sense of tenure security of farm households?
b. Does land registration improve the holding rights of women and marginalized groups in the study area?
c. Is there a change of crop productivity after the land registration process?
d. Does land registration improve perceived tenure security. If yes, what are the influencing factors?
e. Does land registration improve land-management practices such as water-harvesting system. If yes, what are the influencing factors?

### 1.2. Conceptual Framework

Figure 1 depicts a conceptual framework for contribution of land registration. The federal and regional laws provide the foundation for land reform in the form of land registration. The land administrations that have been established are in charge of implementation, which is also dependent on donor support and budget allocations for the activities. The effect of land registration on perceived tenure security, farmer confidence, marginalized groups, and sustainable land-management practice is also influenced by the initial conditions in farming households where reforms are implemented. The effects will be determined by factors such as individual and collectively owned resources and capabilities of households and communities, traditional norms, market exposure, other government policies, and agro-climatic conditions.

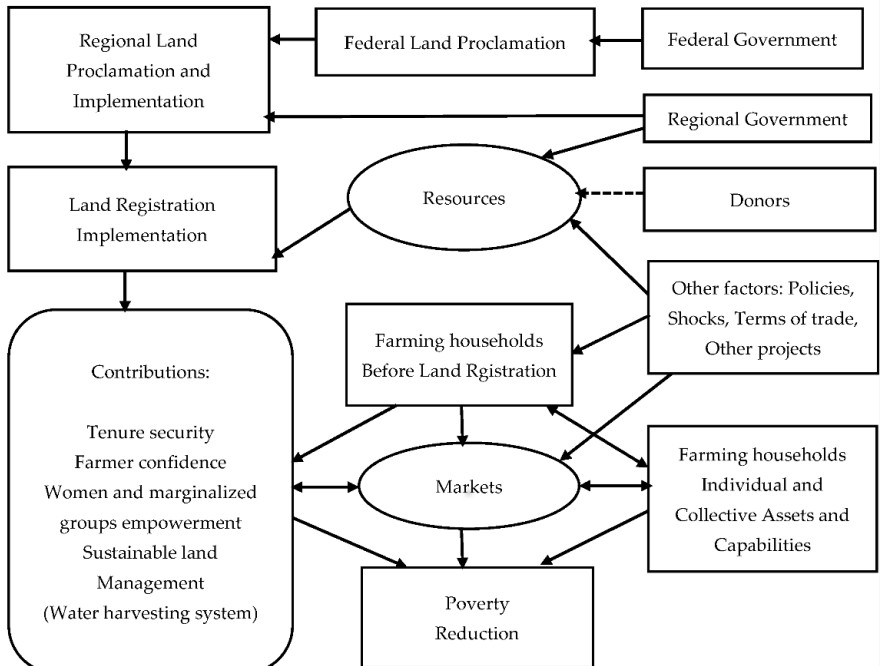

**Figure 1.** Conceptual Framework.

## 2. Materials and Methods

### 2.1. Study Area

East Gojjam Zone is one of eleven zones in Amhara National Regional State, Ethiopia. It is divided into 20 districts, 16 of which are rural and 4 of which are town administration districts. The Zone encompasses an area of approximately 14,004.47 square kilometers. The Oromia Region borders it on the south, West Gojjam Zone on the west, South Gondar Zone on the north, and South Wollo Zone on the east. The bend of the Abay River defines the Zone's northern, eastern, and southern boundaries. Mount Choke (also known as Mount Birhan) is its highest point, rising around 4100 m above mean sea level. East Gojjam Zone stretches from latitude 9°55′01″ to 11°14′12″ north and from longitude 37°29′37″ to 38°30′18″ east (see Figure 2).

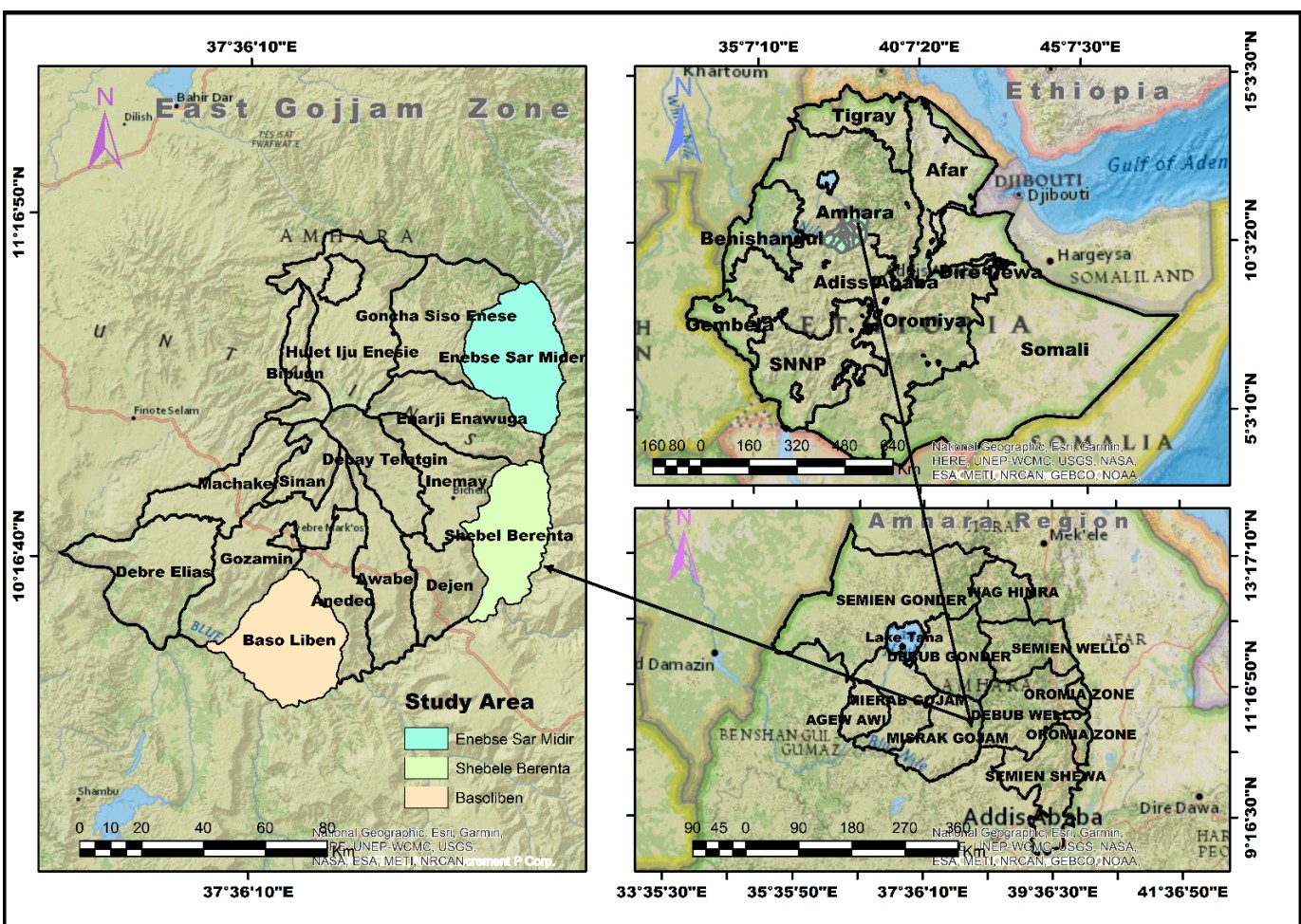

**Figure 2.** Study area map.

East Gojjam Zone is characterized by different landscapes such as mountains (Choke Mountain and Aba Mentous Mountain), plateaus (Yetnora, Awabal and Anaded, Gozamin, Debre Elias) and Gorges (Abay Gorge and Wamet). The study area is located between 759 and 4100 m above sea level. Different vegetation types have resulted from topographic variations combined with diverse climatic conditions, ranging from Afroalpine and sub-afroalpine vegetations to dry evergreen Montane Forest and Combretum Terminalia Woodland. The total population of the East Gojjam Zone is 2,153,937, of whom 1,066,716 are male and 1,087,221 are female. This zone has a population density of 153.80 people per square kilometer and the urban population accounts for 213,568 (9.92%) of the total population, while the remaining 1,940,369 (90.08%) are rural residents. This zone has a total of 506,520 households, with an average of 4.25 people per household and 492,486 housing

units [73]. Because of the implementation of the land certification program, three representative districts, Enebse Sar Midir, Shebele Berenta, and Basoliben, were chosen to collect study data (Figure 2).

*2.2. Methodology*

2.2.1. Sampling Techniques

Three representative Districts (Enebse Sar Midir, Shebele Berenta and Basoliben) of Eastern Gojjam Zone were purposively selected for this study, due to land registration implementation history and dryland areas. The kebeles (lowest administrative structure) with the highest proportion of households registered were identified first in each sample district. Five kebeles were then chosen at random from these, and in total fifteen kebeles were selected from the three districts. With the assistance of kebele managers, administrators, and chairpersons of land administration committees, the names of female, male, and jointly certified households were identified on a separate slip of paper.

All 15,082 total households listed in fifteen selected kebeles were the sampling frame (N) (Table 1). To calculate the sample size (n), the statistical formula Cochran (1977) [74] was used. With a 95% confidence level and a 5% sampling error, the sample size (n) was calculated. As a result, the study's sample size (n) was 385 households. Probability proportional to size principle was used to assign a sample respondent from each kebele (Table 1). Finally, based on the number of respondents assigned to each sample category, the actual sample size was determined using a simple random sampling method. The detailed information about the sample districts, kebeles and sample size taken from each kebele is documented in Table 1.

**Table 1.** Total population of the sample kebeles and total sample size.

| District | Kebeles | Total Population | | | | Sample Size | | | |
|---|---|---|---|---|---|---|---|---|---|
| | | Joint | Male | Female | Total | Joint | Male | Female | Total |
| Enebse Sar Midir | Gesese | 695 | 156 | 336 | 1187 | 18 | 4 | 8 | 30 |
| | Yetefet | 479 | 124 | 160 | 763 | 12 | 3 | 4 | 19 |
| | Ambalaye | 430 | 111 | 226 | 767 | 11 | 3 | 6 | 20 |
| | Segenet | 795 | 336 | 268 | 1399 | 20 | 9 | 7 | 36 |
| | Leule | 666 | 158 | 289 | 1113 | 17 | 4 | 7 | 28 |
| Shebele Berenta | Qarema | 682 | 278 | 424 | 1384 | 17 | 7 | 11 | 35 |
| | Abera | 626 | 324 | 363 | 1313 | 16 | 8 | 9 | 33 |
| | Beneyana Seqela | 789 | 257 | 456 | 1502 | 20 | 7 | 11 | 38 |
| | Gebsit | 575 | 209 | 347 | 1131 | 15 | 5 | 9 | 29 |
| | Yejuna Bayelie | 548 | 293 | 351 | 1192 | 14 | 7 | 9 | 30 |
| Basoliben | Korke | 721 | 139 | 131 | 991 | 18 | 4 | 3 | 25 |
| | Yeduge | 343 | 57 | 88 | 488 | 9 | 1 | 2 | 12 |
| | Anejeme | 427 | 86 | 166 | 679 | 11 | 2 | 4 | 17 |
| | Yelaminje | 392 | 46 | 52 | 490 | 10 | 1 | 1 | 12 |
| | Dendo | 502 | 84 | 97 | 683 | 13 | 2 | 2 | 17 |
| Total | | 8670 | 2658 | 3754 | 15,082 | 221 | 68 | 96 | 385 |

2.2.2. Data Collection Technique

Household surveys (HHS) conducted from September 2022 to October 2022 were the primary source of data. To collect primary data for the field interviews, both closed and open-ended structured questionnaires were used. Structured questionnaires were developed, tested, and adjusted to fit their intended purpose. Farmers were asked before and after land registration about their perceptions of land holding rights and land management activities. Four data collectors with a minimum of bachelor's degree (Undergraduate) in related fields of land administration and land management were employed for data collection. These enumerators were first trained in data collection techniques, study objectives, questionnaire management, and interviewing techniques. Face-to-face interviews were

required because many of the respondents were illiterate. To avoid language barriers, one expert in land administration and management translated the questions from English into Amharic (local language). Official supporting letters written by each district office to the kebeles helped to enable data collection at kebele level.

The questionnaire was designed to gather information about respondents' personal and socioeconomic characteristics, as well as the effect of land registration on sustainable land management. Furthermore, each question was thoroughly explained and clarified to them with adequate explanation. The questionnaire was pre-tested by administering it to selected respondents at Korke kebele. On the fourth day of the exercise, enumerators were given the opportunity to make suggestions and remarks that could help them handle the interview. Based on the results obtained from the pre-test, necessary modifications were made to the questionnaire. Variables identified in the survey are documented in Table 2.

**Table 2.** Variables used in the study.

| Variables | Definition and Values Used |
|---|---|
| Female | Female respondent (=1 female, =0 otherwise) |
| Male | Male respondent (=1 male, =0 otherwise) |
| Joint | Joint (Male and Female) respondent (=1 male and female jointly, =0 otherwise) |
| Age | Age of the respondents (=0, 1, 2, 3, . . . , n) |
| Education | Educational level of the respondents (=1 literate, =0 illiterate) |
| Household size | Total household size (=0, 1, 2, 3, . . . , n) |
| Land holding size | Total land holding size in hectare (=0, 1, 2, 3, . . . , n) |
| Distance | Average distance farm to settlement in minutes (walking) (=0, 1, 2, 3, . . . , n) |
| Land redistribution | Affected by land redistribution of 1997 (=1 yes, =0 no) |
| Expropriation | Fear of loss of land due to the expropriation by government at any time (=1 yes, =0 no) |
| Perceived tenure security | Fear of loss of farmland due to redistribution within the next five years (=1 (do not fear) yes, =0 otherwise) |
| Credit beneficiary | Credit beneficiary of the respondents (=1 yes, =0 no) |
| Training and advice | Training and advice on land management (=1 yes, =0 no) |
| Livestock holding | (Total livestock holding size) =0, 1, 2, 3, . . . , n (in tropical livestock units) |
| Land-management practice | Application of land management (at least one) practices of parcel (=1 yes, =0 no) |
| Water harvesting construction | Application of water harvesting construction (=1 yes, =0 no) |
| Crop yields | Crop yields for major crop types (=0, 1, 2, 3, . . . , n in quintal) |

Data from household surveys were supplemented with qualitative data from direct field observations, focus group discussions (FGD), and key informant discussions. To supplement the quantitative data, FGD were held in each kebele. The FGD participants were chosen based on their knowledge of and experience with land-management practices. These people have lived in the kebeles for a long time and formed the kebele's Land Administration and Certification Committee (LACC). LACC members include elders, female-headed households, youth, and disabled people, as well as Development Agents and Kebele Managers. There were nine group discussions (three in each kebele). With the help of the "Kebele Land Administration Officer", each FGD had 10 to 12 participants. The focus of the discussion was on local-level entities dealing with land-related issues, the effect of land certification on land management, and other issues.

Youth, women, and the elderly were among the community representatives chosen for focus group discussions. A few members of the kebele Land Administration and Certification Committee, development agents, kebele leaders, and district experts were among the key informants. Interviews with representatives of the Environmental Protection and Land Administration Authority, the Bureau of Agriculture and Rural Development, and local authorities were conducted to better understand experts' perceptions of land registration and its intended objectives. Secondary data were gathered by reviewing several reports at the kebele, district, zonal, and regional levels. In addition, five federal, six regional, nine zonal, and four district experts participated in panels and discussion forums. These professionals work in rural land administration and land management offices, as well as other related fields. The professionals' discussion focused on accomplishments, bottlenecks, and recommendations for sustainable land management.

### 2.2.3. Data Analysis Techniques

Most of the data were analyzed quantitatively, and the analysis was supplemented by a qualitative analysis. Descriptive statistics such as percentages, mean, standard deviation, chi-square, and t-test were used to describe the socioeconomic characteristics of respondents. For a more detailed data analysis, a binary logit regression model was used. To determine the effect of other factors on selected variables, the following formula was used:

$$\ln\left[\frac{P_x}{(1 - P_x)}\right] = \sum_{i=1}^{n} \beta_i X_i + U_i$$

where $Px$ is the probability for an observed set of variables that the event occurs, $\beta_i$ is the $i$th coefficient to be estimated, $X_i$ is the $i$th explanatory variable and $U_i$ is a random error term.

### 2.3. Model Specification

For selected discrete and continuous variables, the presence of multi-collinearity and association was investigated. To identify multi-collinearity between continuous variables, the variance inflation factor (VIF) method was used [75]. A contingency coefficient test was used to evaluate associations between dummy variables. The variables found to be highly correlated with one or more of the other continuous or discrete variables (VIF > 10) were excluded from further analysis.

### 2.4. Description of Dependent Variables

Perceived tenure security, a dependent variable, shows whether respondents anticipate losing farmland because of redistribution within the next five years. For respondents who do not expect future redistribution, this binary variable has a value of 1, and for those who do, it has a value of 0. To investigate the influence of land registration on perceived tenure security, the following model was used:

Perceived tenure security = β0 + β1FEMALE + β2JOINT + β3EDU + β4AGE + β5HHSIZE + β6LANDHOLD + β7LANDRED + β8EXPROPRIATION + β9LH + Ui

where β0 to β9 is the coefficient to be estimated, FEMALE, JOINT, EDU, AGE, HHSIZE, LANDHOLD, LANDRED, EXPROPRIATION, and LH are explanatory variables, and Ui is a random error term.

Water harvesting construction, a dependent variable, has a value of 1 if the plot received water-harvesting system application, and 0 otherwise. The explanatory variables, on the other hand, are either continuous or binary. The influence of land registration on the water-harvesting system on plot j by household i was specified as follows:

WHC = β0 + β1FEMALE + β2JOINT + β3EDU + β4AGE + β5LANDHOLD + β6DIST + β8CREDBENEF + β9TRAINING + β10LH + Ui

where β0 to β10 is the coefficient to be estimated, FEMALE, JOINT, EDU, AGE, LAND-HOLD, DIST, CREDBENEF, TRAINING, and LH are explanatory variables and Ui is a random error term.

## 3. Results

### 3.1. Household Characteristics

Land registered in the names of females, males, and joint (male and female) accounted for 23%, 17%, and 60% of the sampled households, respectively. Concerning the educational attainment, 36% of the households sampled were illiterate, 42% could read and/or write, and 22% had completed grade five. Table 3 contains detailed information about the household characteristics.

**Table 3.** Land holding right and education of households.

| Variables | Number | Percent |
|---|---|---|
| Land holding right | | |
| Female | 89 | 23.0 |
| Male | 64 | 17.0 |
| Joint | 232 | 60.0 |
| Total | 385 | 100.0 |
| Educational level | | |
| Illiterate | 140 | 36.0 |
| Read and/or write | 162 | 42.0 |
| Grade five and above | 83 | 22.0 |
| Total | 385 | 100.0 |

Amhara Land Administration and Use (ALAU) Proclamation No. 133/2005 under Article 24 (2) states that where the land is a holding of a husband and a wife in common, the holding certificate shall be prepared by the name of both spouses [70]. As a result, joint titled implies implementation of land proclamation.

The age structure of households revealed that the mean age was 47 years. Furthermore, the average household size was 6.2 persons. Looking at the differences between respondent households, the largest family size was 12 and the smallest was one. There was an average difference of 0.03 ha in land holding size before and after land registration (see Table 4). The average difference between the number of farm plots before and after land registration was only 0.31.

**Table 4.** Summary of age of the household head, household size and land holdings (N = 385).

| Variables | Mean Difference | St. Difference |
|---|---|---|
| Age | 47 | 11.09 |
| Household size | 6.2 | 2.35 |
| Landholding before and after land registration (in ha) | 0.03 | 0.22 |
| Plot number before and after land registration | 0.31 | 0.38 |

### 3.2. Characteristic of Farm Plot

Farm plot characterization was assumed to demonstrate differences in the fertility status of farm plots before and after registration. Based on farmer perception, farms' fertility status could be classified as fertile, moderately fertile, or poorly fertile.

The findings in Table 5 clearly show that land registration had no effect on the level of farmland fertility. Out of all the households surveyed, 258 have an average of 0.60 ha of farmland on a flat slope, 126 have an average of 0.36 ha on a moderate slope, and 163 have an average of 0.40 ha on a steep slope. The cultivation of crops on steep slopes suggests that the study area lacks land use planning and a consequence of demand for land and food production exceeding supply of suitable land, pushing farmers to use marginal land.

**Table 5.** Fertility status of farm plots of sample households.

| Farm Status | Frequency | Before Registration | | After Registration | | |
| | Number (%) | Mean | St. dev. | Number (%) | Mean | St. dev. |
|---|---|---|---|---|---|---|
| Fertile | 266 (88) | 0.59 | 0.52 | 272 (90) | 0.60 | 0.53 |
| Moderate | 193 (63) | 0.44 | 0.34 | 194 (64) | 0.45 | 0.35 |
| Poor | 131 (43) | 0.37 | 0.26 | 127 (41) | 0.39 | 0.27 |

*3.3. Households' Confidence on Land Registration*

According to Table 6, the last land redistribution affected 23% of the sampled households interviewed, either positively or negatively. However, approximately 26% of households are concerned about land redistribution over the next five years and losing their farms. Furthermore, 22% of households are concerned that the government will seize their farm plot at any time. Focus group participants proved that at each kebele, farmers living around town administration were highly frustrated with the expropriation of their farms.

**Table 6.** Households' confidence on land redistribution after registration (N = 385).

| Name of the Variable | Yes | | No | |
| | Number | Percent | Number | Percent |
|---|---|---|---|---|
| Households affected in 1997 land redistribution | 90 | 23 | 295 | 77 |
| Fear of land redistribution and farm loss in the next five years | 102 | 26 | 283 | 74 |
| Fear of government land expropriation at any time | 83 | 22 | 302 | 78 |

*3.4. The Effect of Land Registration on Women and Marginalized Groups*

Table 7 shows that approximately 70% and 85% of households knew landless households in their village before and after land registration, respectively. The statistical test reveals a significant ($p < 0.01$) difference between the number of landless households in marginalized social groups before and after land registration. The qualitative information gathered from household surveys and focus group discussions showed that land registration protects the land rights of women and other marginal societies more than youths.

**Table 7.** Household Landlessness before and after land registration (N = 385).

| Variable Name | Before Land Registration (Percent) | | After Land Registration (Percent) | | Chi-Square |
| | Yes | No | Yes | No | |
|---|---|---|---|---|---|
| Landless households | 70.0 | 30.0 | 85.0 | 15.0 | 1.37 *** |
| Women | 52.7 | 47.3 | 37.0 | 63.0 | 94.73 *** |
| Disabled | 41.7 | 58.3 | 30.0 | 70.0 | 1.21 *** |
| Youth | 60.3 | 39.7 | 84.7 | 15.3 | 63.57 *** |
| Poorest of poor | 51.7 | 48.3 | 30.3 | 69.7 | 79.06 *** |

*** = Significant at $p < 0.01$.

The chi-square test reveals a significant difference ($p < 0.01$) before and after land registration in the case of women's stronger land-holding rights in jointly led households. Furthermore, according to focus group participants and key informants, women have full rights to share the land equally during divorce; no one takes the land of women and other marginal societies.

### 3.5. Effect of Land Registration on Land-Management Practices

Table 8 clearly demonstrates that, with $p < 0.01$, approximately 80% and 95% of the households participated in at least one type of land management practice before and after land registration, respectively. Land management practices considered in this study included terracing, tree planting, compost application, manure application and the construction of water-harvesting structures (WHS). There was a significant difference between before and after land registration for each type of land management practice ($p < 0.01$).

**Table 8.** Land management practices in (%).

| Factors | Before Land Registration | | After Land Registration | | Chi-Square |
|---|---|---|---|---|---|
| | **Yes** | **No** | **Yes** | **No** | |
| Land-management application | 80 | 20 | 95 | 5 | 47.65 *** |
| Terracing | 75.0 | 25.0 | 92.3 | 7.7 | 62.89 *** |
| Planting of tree | 45.3 | 54.7 | 50.7 | 49.3 | 2.89 *** |
| Compost use | 40.3 | 59.7 | 65.0 | 35.0 | 2.24 *** |
| Manure use | 70.3 | 29.7 | 83.0 | 17.0 | 1.34 *** |
| Water-harvesting structure | 15.7 | 84.3 | 25.3 | 74.7 | 97.73 *** |

*** = significant at $p < 0.01$.

### 3.6. Effect of Land Registration on Crop Productivity

Except for wheat, there was no significant difference in major crop yields between 2021/22 (after land registration) and 2005/06 (before land registration) ($p < 0.1$). The average difference between wheat production in 2021/22 and 2005/06 is 0.89 quintals per household (Table 9). This finding indicates that there is no significant improvement in major crop yield per household following land registration, but rather a decrease. This could be due to changes in rainfall and other factors.

**Table 9.** Major crops produced in the year 2021/22 and 2005/06 (quintal/household).

| Crops | Respondents | Difference in Mean | Difference in Std. Deviation | *t*-Test |
|---|---|---|---|---|
| Maize produced in (2021/22–2005/06) | 66 | 0.51 | 3.74 | −1.22 |
| Wheat produced in (2021/22–2005/06) | 182 | 0.89 | 7.15 | 1.74 * |
| Teff produced in (2021/22–2005/06) | 150 | 0.31 | 4.33 | 1.26 |
| Other crops produced in (2021/22–2005/06) | 125 | 0.18 | 5.15 | 0.28 |

* = significant at $p < 0.1$.

### 3.7. Results of the Logit Model

To evaluate the relationships between dummy variables, a contingency coefficient test was used. The model is included for analysis because, as is evident from the results in Appendix A, there are no problems with multi-collinearity between the variables, and the contingency coefficient test result is very good.

### 3.7.1. Influencing Factors of Land Tenure Security

Three of the eight independent variables entered the model, namely education (significant at $p < 0.05$), land holding size (significant at $p < 0.01$), and land redistribution (significant at $p < 0.01$), were significantly and positively influencing households' fear of future land redistribution and loss of farmland (see Table 10).

**Table 10.** Factors influencing land registration on perceived tenure security (N = 385).

| Explanatory Variables Name | B | Z-Value | $p > |Z|$ | Marginal Effect |
|---|---|---|---|---|
| Female | −0.4596 | −0.87 | 0.324 | −0.0887 |
| Joint | −0.0850 | −0.13 | 0.719 | −0.0136 |
| Age | −0.0121 | −0.28 | 0.755 | −0.0015 |
| Education | 0.2562 | 2.02 ** | 0.035 | 0.0649 |
| Household Size | −0.0257 | −0.23 | 0.714 | −0.0138 |
| Land Holding size | 0.6740 | 2.83 *** | 0.004 | 0.1647 |
| Land Redistribution | 3.6758 | 7.14 *** | 0.000 | 0.9216 |
| Livestock holding | 0.0081 | 0.22 | 0.732 | 0.0122 |
| Constant | −2.3673 | −3.43 | 0.001 | |
| Log likeihood | | | −65.42575 | |
| Chi squared | | | 174.52 | |
| Pseudo R$^2$ | | | 0.5592 | |

*** and ** designate significance at $p < 1\%$ and $p < 5\%$, respectively: B (coefficients).

### 3.7.2. Factors Influencing Construction of Water-Harvesting Systems

Before land registration, as shown in Table 11, household size, livestock holding, and distance all had a significant effect on construction of water-harvesting systems at $p < 0.05$, $p < 0.01$, and $p < 0.01$, respectively. After land registration, age and household size influenced the construction of water-harvesting system with a significant difference of $p < 0.05$, whereas distance, livestock holding, and training and advice influenced the construction of water-harvesting systems with a significant difference of $p < 0.01$. The descriptive statistics also revealed that after land registration, the construction of water-harvesting systems increased by 59%.

**Table 11.** Influencing factors of land registration on construction of water-harvesting systems (WHS) (N = 385).

| Variables | Before Land Registration | | | After Land Registration | | |
|---|---|---|---|---|---|---|
| | B | Z-Value | Marginal Effect | B | Z-Value | Marginal Effect |
| Female | −0.3220 | −0.74 | −0.0523 | −0.1490 | −0.51 | −0.0474 |
| Joint | −0.3724 | −1.27 | −0.0740 | −0.3360 | −1.24 | −0.1077 |
| Age | −0.0235 | −1.34 | −0.0015 | −0.0156 | −2.07 ** | −0.0051 |
| Education | −0.1213 | −1.13 | −0.0216 | −0.1285 | −1.35 | −0.0411 |
| Household size | 0.1180 | 2.27 ** | 0.0221 | 0.0892 | 2.01 ** | 0.0276 |
| Land holding size | −0.0674 | −0.31 | −0.0135 | −0.2760 | −1.34 | −0.0758 |
| Training and advice | 0.2694 | 0.78 | 0.0442 | 0.8953 | 2.69 *** | 0.2089 |
| Livestock holding | 0.0987 | 2.50 *** | 0.0183 | 0.1127 | 2.68 *** | 0.0349 |
| Credit beneficiary before LC | −0.1124 | −0.52 | −0.0176 | - | - | - |
| Credit beneficiary after LC | - | - | - | 0.1214 | 0.61 | 0.0479 |
| Distance | −0.0075 | −2.49 *** | −0.0014 | −0.0078 | −3.33 *** | −0.0027 |
| Constant | −0.9554 | −1.55 | | −0.5520 | −1.04 | |
| Log likelihood | | −104.6043 | | | −156.23964 | |
| Chi squared | | 25.30 | | | 44.47 | |
| Pseudo R$^2$ | | 0.1025 | | | 0.1126 | |

*** and ** indicate significance at $p < 1\%$ and $p < 5\%$, respectively: B (coefficients).

## 4. Discussion

### 4.1. Confidence of Households in Land Tenure Security

The last land redistribution affected 23% of the households in 1997 (Table 6). According to Deininger et al. [76], land redistribution affected 9% of Ethiopian farmers between 1991 and 1998, 18% in Tigray region, and 21% in the Amhara. Approximately 26% of households are concerned about land redistribution and losing their farm plots in the next five years, while 22% are concerned that the government will take their farms at any time (see Table 6). Thus, there is still concern about land redistribution, as found in Tigray, where 44% of farmers expect land redistribution and believe they will lose farms [77].

Furthermore, a previous study has found that 27% of respondents are confident that land redistribution will not happen in the future, while 9% believe it will occur within the next five years [78]. Given the aim of land certification, a small number of households are concerned about land redistribution over the next five years, and the government must address those households properly if land management is to improve.

### 4.2. The Effect of Land Registration on Women and Marginalized Groups

Land registration aimed to protect the land rights of marginalized groups such as the elderly, disabled, and women. After land registration, women, the disabled, and the poorest of the poor experienced less landlessness, whereas youths experienced an increase. This result was consistent with previous studies and discovered that 8.5% of farm holders are younger than 24 years old, indicating that landlessness is a significant issue in the Amhara region, especially for young people who have difficulty accessing land. This could be due to a lack of farmland, and land law prioritizes youths as one of society's most marginalized groups [61]. For instance, revised Amhara Region Land Administration and Use (ARLAU) Proclamation No. 133/2006 Article 9 (2) supports land holding in priority order for orphans, the disabled, women, and young people who join the new life of independence.

Women now have more land ownership rights after receiving land certification. Land registration has been shown in studies to promote gender equality, increase women's tenure security, and enhance land-management practice participation; [66,79,80] supported this conclusion. According to similar studies, the land certificates promote gender equality and encourage women to the field work [79]. Furthermore, this finding is in line with results found in Amhara Region pilot and non-pilot districts [81] and in Southern Ethiopia, who discovered that certification improved women's tenure security [82]. According to studies, the majority of households (85%) believe that land certification will improve women's status and provide incentives for land rental [66]. Finally, the land registration program promoted gender equality in Worja kebele in the Southern Nations Nationalities and Peoples region and 90% in Beresa kebele in Oromia [80].

### 4.3. Land Registration Effect on Land-Management Practices and Crop Productivity

Following land registration, household participation in land-management practices improved (see Table 8). This result is consistent with research carried out in Tigray, where 85.2% of households engaged in various sustainable land-management practices following land titling, compared to 34.1% growth prior to titling [83]. Similar studies discovered that a sizable majority of households in Ethiopia believed that registration of rural land increased incentives for spending on planting trees (88%), building structures for soil and water conservation (86%), and managing common property resources sustainably (66%) [66]. Likewise, land registration has strong implications for household participation in sustainable land management initiatives at the community level [84].

Most of the land tenure regularization programs predict an increase in land-based investments such as soil and land management infrastructure due to land registration and certification [85]. Deininger et al. [29] found positive and a statistically significant marginal effect of the land certification on the repairs and new investments in land management with an estimated average treatment effect of 30%. Land management incentives promote the positive impact of the land registration program [86]. In order to increase investments in

land-related projects for sustainable land management, certificates are issued [29]. According to a similar report, 77.5% of Worja kebele farmers in the Southern Nation Nationalities and Peoples region and 70% of Beresa kebele farmers in Oromia completely agree that land registration increases investments in soil and land management [80]. In the same manner, reports indicate that 96.7% of farmers in pilot areas and 77.5% of farmers in non-pilot areas in the Amhara have participated in land management activities [81]. Additionally, studies conducted in Damot-Gale District, Southern Ethiopia revealed that the majority (62%) of the respondents indicated that they are practising land management due to a certificate, i.e., land certificate increases the perception of farmers in land management practices [87]. Finally, results from Melesse and Bulte [56] substantiate that land-certified households are more likely to adopt land management strategies than the uncertified ones. The participants of focus group discussion clearly indicated that land registration addressed issues of persistent gender inequality. As a result, registration improved decision-making in relation to land-management practices, and increased women's land rights. Studies show that the registration process made women more willing to work in the field and apply appropriate land-management practices [80].

Major crop yields decreased following land registration, except for wheat. The average difference in wheat production before and after land registration per household decreased by 0.89 quintals. This result demonstrates that, rather than improving significantly after land registration, major crop yield per household decreased. This might be brought on by changes in rainfall and other factors affecting crop growth. Because frequent droughts, the recent emergence of insect pests, and other factors have an impact on farmland productivity, crop productivity did not increase solely because of land registration in dryland areas. This outcome is consistent with earlier findings, according to which 50% of households in non-pilot districts and 63.3% of households in pilot districts of the Amhara region both agreed that land registration had no impact on farmland productivity [81]. On the contrary, studies have shown that improved land-management practices following registration have been associated with increased crop yields [71].

### 4.4. Factors Affecting Perceived Tenure Security

### 4.4.1. Education

The educational level of the respondents has a significant and positive effect on the fear of future land redistribution (see Table 10). Respondents who have higher levels of education are more likely to engage in off-farm activities and find alternative employment opportunities. As a result, there is a greater fear of losing farmland because the government could take over the land at any time. This survey result is consistent with the Amhara Rural Land Administration and Use (ARLAU) Proclamation No. 133/2006, which states in Article 12 (1a) that any land holder of a right to use the land may lose that right if he engages in non-farming activities and makes a living from these [70]. As a result, households are concerned that as education levels rise, so will the likelihood of non-agricultural activity, which may not be enough to meet individuals' basic needs but will result in farmland loss. On the contrary, this finding contradicts Pender et al. [88], in which the findings reported that education is likely to increase households' opportunities for salary employment off-farm and may increase their ability to start up various nonfarm activities. In addition, this may increase households' access to credit as well as their cash income, thus helping to finance purchases of physical capital and purchased inputs.

### 4.4.2. Landholding Size

There is a positive and significant correlation between respondents' total land holding size and their fear of future land redistribution (Table 10). Households with large land holdings are more concerned about land redistribution and the loss of farmland. According to participants in the focus group discussion, as food insecurity and crime rise, an increasing number of landless young people are threatening their farms. Consequently, land could be redistributed from elderly people who own large farms to landless youth. The findings

of this study are supported by investigations that farm households with relatively larger farms feel more insecure than those with relatively less land, and farm tenure security in Ethiopia is inversely related to farm size [71,89].

### 4.4.3. Land Redistribution

The fear of land redistribution is significantly and positively associated with households affected by land redistribution in 1997 (Table 10). Fear of land redistribution is high, and it is even higher within the next five years than it is beyond (Table 5). Focus group discussion participants reported that land redistribution had occurred frequently in recent days. They have no idea what will happen in the future because it is dependent on the government and its policies. Even the land policy gives reason for concern, stating that land redistribution may be possible if the land is required for irrigation projects. Another concern is that the government could be replaced, and the legislation would not be properly implemented [71].

### 4.5. Factors Influencing Construction of Water-Harvesting System

#### 4.5.1. Distance

The construction of a water-harvesting structure is negatively impacted by the distance between a farm plot and the settlement. After land registration, the average distance of a farm plot from the settlement increased by one minute, while the construction of a water-harvesting structure decreased by 0.27%. Farmland owners who live close to residential areas are more likely to build a water-harvesting system than those who live far from the settlement (see Table 11). This is since households prefer nearby farm plots over distant plots. According to studies, managing close farmland takes less time and energy, so longer walking distances between farmland and settlement areas reduce farmland cultivation adoption [90,91].

#### 4.5.2. Training and Advice

The construction of a water-harvesting system has a significant and positive relationship with households that received land-management training and advice (see Table 11). A unit increase in training and advice from agricultural extension services increased the construction of a water-harvesting structure by 20.8% after the land was registered. This significant increase was caused by the provision of enough knowledge and instruction on sustainable land-management techniques. Focus group participants reported that households that received more training were motivated to build water-harvesting systems for their farmlands. Participants acknowledged that extension services offered to them were more likely to persuade them to make such land investments than development workers. Previous research has found that farmers who receive training are more likely to adopt, use, and implement land-management practices [91]. Farmers' attitudes and abilities in land management will improve because of increased access to training, as well as their knowing of the advantages and limitations of soil conservation. Additionally, training enhances one's capacity to understand and use specific knowledge about land management activities. A previous study confirmed that training had an impact on the adoption of land-management applications [90–93]. Numerous studies have examined the connection between farmers' training and their use of sustainable land-management techniques [94]. The current study's findings also showed that after the land registration process, the impact of training was increased.

#### 4.5.3. Livestock Holding

The total number of livestock holdings and the decision to construct a water-harvesting system were significantly and positively correlated (see Table 11). Following land registration, the construction of water-harvesting systems increased by 3.4% as the total number of livestock increased by one unit. Small family sizes, a labor shortage, and a high livestock population are the most likely causes. There is a chance of selling livestock and converting

to human labor. It is then possible to build water-harvesting systems using the human labor force gained from family members. Key informant participants confirmed that when farm households have a shortage of human labor, livestock sales are used to purchase labor for the building of water-harvesting systems. The results of this study have also been supported by earlier studies [90,93,95,96]. Additionally, livestock is a significant source of farm income that enables farmers to invest in land management strategies and purchase agricultural inputs. Moreover, it serves as non-human labor to construct structures for soil conservation [97]. According to earlier research, the quantity of livestock is a sign of financial stability, which improves the efficiency of land management [95,96].

### 4.5.4. Household Size

After land registration, the size of a household has a positive and significant influence on the construction of a water-harvesting structure. Construction of water-harvesting structures increased by 2.7% as household size increased by one member after certification (Table 11).

### 4.5.5. Age

After land registration, the construction of a water-harvesting structure is negatively impacted by the household head's age. As a result, after land registration, an increase of one year in the household's age resulted in a 0.51% decrease in the construction of water-harvesting structures (Table 11). Older farmers have larger land holdings than younger farmers, and they may lose land due to redistribution. So, older farmers were less invested in land management [98]. Greater family labor indicates a greater potential for labor-intensive investments such as water-harvesting construction. Larger households will be able to provide the labor needed to maintain conservation structures [98].

## 5. Conclusions and Recommendations

Land registrations are critical issues in Ethiopia's land administration system for improving land tenure security. To that end, Ethiopia has had a land registration and certification program in place since 1998, and the Amhara region has had one since 2002, with the goal of registering all land holdings and issuing land certificates to enhance farmers' land rights' security. In theory, land certification stimulates economic growth by providing incentives to increase agricultural production. Secure land rights are essential for economic development.

Because tenure insecurity is a problem in African countries, efforts should be made to provide land rights to people, and particularly to women and marginal groups. Appropriate land rights are considered a starting point for the empowerment of the poor. Land registrations are currently applied in Ethiopia to provide land tenure security. This contributes to the advancement of sustainable land-management practices. As a result, developing countries can learn from this success and emphasize tenure rights for their country's sustainable development.

Even though land registration has a significant impact on long-term land management, Ethiopia lacks a comprehensive land use policy. Land use regulation is not given much weight in the current rural land administration system. Land use rights are given less attention in rural land administration and land use proclamations. The legal framework is primarily concerned with issues of land administration. Of course, land ownership and tenure security are fundamental components of sustainable land-management practices, and they are a good place to begin. Nonetheless, in order to enforce sustainable land management, a land use policy for proper land use practices should be established. Otherwise, there will be no solution to the land degradation and deforestation problem. This, in turn, could be a threat for agricultural production and exacerbate the country's poverty situation.

In this study, land registration significantly improved farmers' perceptions and confidence in land tenure security, even though 13.7% of households remain concerned about future land redistribution and expropriation by the government at any time. Fear of land

expropriation by the government emerged from the foundation of new town administration and that is holding rural kebeles.

The state still owns all land in Ethiopia, even though farmers feel more secure regarding competing claims to their land from neighbors and relatives. This policy continues to create insecurity, especially when local officials suggest that the government might seize the land if it is not used properly. Such claims by local officials have caused confusion among smallholders about the benefits of land registration on tenure security, sparking a debate about whether land registration must be accompanied by land ownership in order to realize secure use rights.

The logit model results revealed that education, land holding size, and households affected by last land redistribution were found to significantly and positively aggravate households' fear of future land redistribution and farmland losses. Except for youths, land registration effectively protects the land use rights of women and other marginalized groups in society. As a result of this, youths have raised the issue of land right immediately following land registration in the study area. Women's land holding rights were found to be stronger after land registration, with a significant difference of $p < 0.01$ between before and after land registration.

Regarding the effect of land registration on land management on cultivated land, household land management participation improved after land registration in the study area. However, the average distance of a farm plot from the settlement had a negative impact on the construction of water-harvesting system in the study area, whereas access to agricultural extension training and advice, as well as livestock holding, had a positive impact on the construction of water-harvesting systems. Nonetheless, steep slope areas in the study area are still used for crop cultivation. With the exception of wheat, which was significant at the $p < 0.1$ level, there was no significant difference in major crop yield per household after and before land registration. This result shows that there is no significant improvement in major crop yield per household after land registration, but rather a decrease. This could be due to variations in rainfall and other crop growth factors. As a result, crop yield did not improve solely through land registration in dryland areas because farmland productivity was affected by the occurrence of recurrent drought and other factors.

The findings also provide important policy implications and suggest that policymakers both at governmental and non-governmental agencies engaged in sustainable land management among rural agricultural households that aim to boost agricultural development should consider land registration as an important prerequisite. The evidence shows that if farm households are given more secure property rights on their land, they would be encouraged to increase their investments in sustainable land management. Thus, policymakers in Ethiopia should consider land registration as a matter of priority to ensure the success of sustainable land management programs and to promote the development of modern agriculture. Tenure security by ensuring the probability of benefiting from their investment in the long term. Thus, tenure security can also serve as an incentive mechanism for the success of sustainable land management.

The possible recommendations were that governmental and non-governmental offices should work together to raise awareness about the duties and responsibilities that land registration entails. Meanwhile, the government should look for clear policies, such as small-scale enterprise and urban agriculture, to address the issues of landless youths and farmers whose lands have been encroached upon by town administration expansions into rural kebeles. Furthermore, the government should strengthen the implementation of the society's land registration processes. However, to address youth landlessness, intensive farming practices should be promoted, which will increase labor needs and thus engage youths.

**Funding:** This research received no external funding.

**Data Availability Statement:** The data presented in this study are available on request from the corresponding author. The data are not publicly available due to restrictions e.g., privacy or ethical.

**Acknowledgments:** The author thanks respondents and land administration experts for their collaboration during survey interviews.

**Conflicts of Interest:** The authors declare no conflict of interest.

## Appendix A

Severity of multi-collinearity between independent variables was assessed prior to estimating the logit model by calculating the variance inflation factor (VIF). VIF < 10 specifies that there is no multi-collinearity.

**Table A1.** Multi-collinearity test for perceived tenure security.

| Continuous Independent Variables | VIF |
|---|---|
| Age | 1.112 |
| Household size | 1.184 |
| Land holding size | 1.158 |
| Livestock holding | 1.165 |

VIF is variance inflation factor; source: survey, 2022.

**Table A2.** Multi-collinearity test for construction of water-harvesting system.

| Explanatory Variables | VIF | |
|---|---|---|
| | Before Registration | After Registration |
| Age | 1.158 | 1.123 |
| Household size | 1.214 | 1.187 |
| Land holding size | 1.159 | 1.151 |
| Distance | 1.022 | 1.021 |
| Livestock holding | 1.143 | 1.135 |

VIF is variance inflation factor; source: survey, 2022.

**Table A3.** Contingency coefficients for perceived tenure security.

| | Female | Joint | Education | Land Redistribution | Expropriation |
|---|---|---|---|---|---|
| Female | 1 | | | | |
| Joint | 0.568 | 1 | | | |
| Education | 0.086 | 0.087 | 1 | | |
| land redistribution | 0.023 | 0.023 | 0.032 | 1 | |
| Expropriation | 0.011 | 0.045 | 0.084 | 0.038 | 1 |

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
