# Peer review of "Analysis of the Contribution of Land Registration to Sustainable Land Management in East Gojjam Zone, Ethiopia"

_land, doi:10.3390/land12061157_

Round 1

Reviewer 1 Report (New Reviewer)

This paper describes a state of the art that often is unknown to the world. It is imperative to publish it; the paper with the additions in yellow has improved a lot. 

In Table 10 female lacks an e

in line 525 the author needs to explain the contradiction. 

I personally don't like to finish saying that more research is needed ("Finally, additional research should be conducted to investigate factors influencing crop productivity"), I mean, it is so general. Throughout the paper, this question is filtered, but I suggest adding or quoting other papers suggesting the hypothesis of drought, crop genetic variants, technology, or whatever; I am sure there are other countries with similar problems that the author can mention. 

Author Response

Response to Reviewer 1 Comments

We would like to express our sincere thanks to the reviewer for the valuable comments/suggestions. Our point-by-point responses are given below in red colour.

Comments and Suggestions for Authors:

This paper describes a state of the art that often is unknown to the world. It is imperative to publish it; the paper with the additions in yellow has improved a lot.

Thank you very much for your appreciation and comments.

Specific comments :

Point 1: In Table 10 female lacks an e

Response 1: Thank you for your comment and correction. According to your correction, I added “e”. See the corrections in Table 10 of the manuscript.

Point 2: in line 525 the author needs to explain the contradiction.

Response 2: Thank you for your comment. I explained the contradiction such that the findings reported that education is likely to increase households’ opportunities for salary employment off-farm and may increase their ability to start up various nonfarm activities. Also, may increase households’ access to credit as well as their cash income, thus helping to finance purchases of physical capital and purchased inputs. See the additions in Lines 526 – 529 in the revised manuscript.

Point 3: I personally don't like to finish saying that more research is needed ("Finally, additional research should be conducted to investigate factors influencing crop productivity"), I mean, it is so general. Throughout the paper, this question is filtered, but I suggest adding or quoting other papers suggesting the hypothesis of drought, crop genetic variants, technology, or whatever; I am sure there are other countries with similar problems that the author can mention.

Response 3: Thanks for your valuable comments and suggestion. So I agreed with your comment and I deleted from the manuscript the recommendation that “Finally, additional research should be conducted to investigate factors influencing crop productivity”. But my intention is the effect of land certification on crop productivity and my result showed that rather than improving significantly after land certification, major crop yield per household decreased. This might be brought on by changes in rainfall and other factors affecting crop growth. Because frequent droughts, the recent emergence of insect pests, and other factors have an impact on farmland productivity, crop productivity did not increase solely because of land certification in dryland areas. This outcome is consistent with earlier findings, according to which 50% of households in non-pilot districts and 63.3% of households in pilot districts of the Amhara region both agreed that land certification had no impact on farmland productivity [80]. On the contrary, studies have shown that improved land management practices following certification have been associated with increased crop yields [70]. See the revisions in lines 692-694 in the revised manuscript.

Reviewer 2 Report (New Reviewer)

Thank you for submitting this excellent paper. I have only a few minor concerns which are detailed in the comments of the attached pdf 9see orange highlights). It was a bit distracting to read a draft with so many yellow highlights - please make sure that you submit a clean version in future.

Highlighting some of the main points for revision:

1. You mention the 'land issue' in Ethiopia 9line 91). This is a loaded term for most post-colonial states and carries different connotations. please describe exactly what you mean. You also refer to land redistribution often, and this also has different connotations depending on the context. Please define it for your context.

2. Lines 293 - 310 appear to be incomplete? it is very difficult to understand what is presented there. Please revise in line with my comments in the pdf.

3. References are made to land-based statistics such as plot size and fertility status. Where do these data come from? Are they perceptions or estimates of (often illiterate) land holders, or are they derived from some official source?

Please attend to these comments and then your paper should be good to go. Well done again on a thorough and interesting piece of research.

Generally, the English is very good. There are a few instances where improvements can be made, but overall these do not detract from the meaning of the text. I have highlighted one or two instances where language misuse may cause confusion.

Author Response

Response to Reviewer 2 Comments

Thank you, dear reviewer, for your valuable comments/suggestions to enrich the paper. We tried to address all issues to improve the paper. Replies to each review comment are given below in red colour.

Comments and Suggestions for Authors:

Thank you for submitting this excellent paper. I have only a few minor concerns which are detailed in the comments of the attached pdf 9see orange highlights). It was a bit distracting to read a draft with so many yellow highlights - please make sure that you submit a clean version in future.

Thank you very much for your appreciation and gave detailed comments in orange highlights in pdf format. Sorry for the inconvenience to read a draft with so many yellow highlights and now I removed the yellow highlights from the manuscript.

Highlighting some of the main points for revision:

Point 1: You mention the 'land issue' in Ethiopia 9line 91). This is a loaded term for most post-colonial states and carries different connotations. please describe exactly what you mean. You also refer to land redistribution often, and this also has different connotations depending on the context. Please define it for your context.

Response 1: Thank you very much for your important comment. So land issue means that land reform and added land reform in line 92. In the Ethiopian case, land redistribution means redistributed land to the tillers. I added in the manuscript the Ethiopian context of land redistribution in lines 95-96.

Point 2: Lines 293 - 310 appear to be incomplete. it is very difficult to understand what is presented there. Please revise in line with my comments in the pdf.

Response 2: Thank you for your comment. In line 293 I corrected according to your comment the sub-title as only ” Description of dependent variables”.

So, the first dependent variable is ”Perceived tenure security”. And shows whether respondents anticipate losing farmland because of redistribution within the next five years. For respondents who do not expect future redistribution, this binary variable has a value of 1, and for those who do, it has a value of 0.

To investigate the influence of land certification on perceived tenure security, the following model was used:

Perceived tenure security = β0 + β1FEMALE + β2JOINT + β3EDU + β4AGE + β5HHSIZE + β6LANDHOLD + β7LANDRED + β8EXPROPRIATION + β9LH + Ui

where β0 to β9 is the coefficient to be estimated, FEMALE, JOINT, EDU, AGE, HHSIZE, LANDHOLD, LANDRED,  EXPROPRIATION, LH are explanatory variables and Ui is a random error term.

The second dependent variable is “water harvesting construction”. And has a value of 1 if the plot received a water harvesting system application, and 0 otherwise. The explanatory variables, on the other hand, are either continuous or binary.

To assess the influence of land certification on the water harvesting system on plot j by household I was specified as follows:

WHC = β0 + β1FEMALE + β2JOINT + β3EDU + β4AGE + β5LANDHOLD + β6DIST + β8CREDBENEF + β9TRAINING + β10LH + Ui

where β0 to β10 is the coefficient to be estimated, FEMALE, JOINT, EDU, AGE, LANDHOLD, DIST, CREDBENEF, TRAINING, and LH are explanatory variables and Ui is a random error term.

For the changes and rewriting see lines 380 to 403 in the revised manuscript.

Point 3: References are made to land-based statistics such as plot size and fertility status. Where do these data come from? Are they perceptions or estimates of (often illiterate) landholders, or are they derived from some official source?

Response 3: Thank you very much for your nice comments. But the study was undertaken by cross-sectional surveys and even though 36% of the respondents were illiterate know the plot size and fertility status of the plot. Why do the landholders know that during registration the attributes of plots such as plot size (in hectares), fertility status of the plot, name of the landholder and others were registered by the land surveyor and told to the farmers who also gave a certificate that consisted of the different attributes? I mentioned the source of data in the manuscript that based on farmer perception, farms' fertility status could be classified as fertile, moderately fertile, or poorly fertile. See the source of data on land-based statistics in lines 341 – 342.

Please attend to these comments and then your paper should be good to go. Well done again on a thorough and interesting piece of research.

I believe I addressed your comments.

Generally, the English are very good. There are a few instances where improvements can be made, but overall these do not detract from the meaning of the text. I have highlighted one or two instances where language misuse may cause confusion.

Thanks, Throughout the manuscript, I attempted to correct and edit the English.

Additional comments in the PDF document:

Line 91 The land issue in Ethiopia is land reform that is after overthrowing the imperial regime of Haile Selassie through a military coup (1974), the socialist Derg regime implemented radical reforms that altered the agrarian structure and access to land, transferring land ownership to the state. Furthermore, land redistribution was primarily carried out in the years immediately following the 1975 governmental change, but additional land redistributions have occurred since then (constitutionally this requires a significant majority to demand a land redistribution to take place) [63]. As a result of these legal changes, and significant land holding shifts, smallholders did not perceive that they had a high degree of land tenure security—the land redistribution after all was only usufruct rights, not ownership rights. This tenure system was largely continued with the entrance of the new government in 1991, which made only minor changes to the ability to rent land on a short-term basis.

land redistribution (where the aim was to redistribute land according to the needs (family size) of households and to provide land for young married couples, women, marginalized groups, and youth).

Line 134: I corrected it with the exception of the term in addition to this.

Line 185 the area is ETHIOPIA, not Yemen. Please see the locational information. The first map is Ethiopia and the second Amhara region, and the third map is the East Gojjam Zone consisting of the three districts.

Line 333: average difference of 0.03 ha in land holding size, this determined by the survey of farming households. Already the plot sizes were surveyed by a land surveyor during registration and told the sizes, fertility status and other parameters to the farmers. The farmers told to me during the household survey in the data collection process.

Line 352: Thank you very much I added your good explanation. See lines 440 to 442 in the revised manuscript.

All in all, I incorporated your comments given in the PDF document.

Thank you very much for your understanding.

Reviewer 3 Report (New Reviewer)

- The parts highlighted in yellow came to me directly from the Journal Editor. I think this yellow highlights must have been inserted by another reviewer who had had worked on the manuscripts before the manuscript was sent to me for review. My own highlights are in brown imposed on the yellow.

- The manuscript falls short in Section One for lack of Research Questions, which should guide the presentation of results in an orderly fashion. Please see Lines 156-158.

- Just before Section 2 (Line 159), it would be better if the author could insert a new section on Conceptual Framework. This new section should be limited to about two paragraphs only. Its purpose is to describe the conceptual underpinnings of the paper, which should also guide data analysis under the research questions.

- Figure 1. comprises three smaller figures. This Figure 1 should be split as Figures 1.1, 1.2 and 1.3. under Section 2.1.

- In order to enhance the analytical rigour of the paper, the whole of Section 3 (Results) must be properly restructured in tandem with the Research Questions listed in Section 1. 

- Your discussion should follow the order in the preceding Section (Results) Here, the author should compare the findings with the findings in relevant previous studies as done in Line 525.

- The author should revisit the Conclusion, which should be a standalone section. Here, we should see a clear summary of the findings under each research question.

- The ubiquitous listing of 'Author Contributions' is very unnecessary because the manuscript has only one author. 

The literary composition of the paper is fairly good, but the author should correct the errors identified.  

Author Response

Response to Reviewer 3 Comments

We would like to express our sincere thanks to the reviewer for the valuable comments/suggestions. Our point-by-point responses are given below in red colour.

Comments and Suggestions for Authors:

The parts highlighted in yellow came to me directly from the Journal Editor. I think these yellow highlights must have been inserted by another reviewer who had worked on the manuscripts before the manuscript was sent to me for review. My own highlights are in brown imposed on the yellow.

Thank you very much for the detailed comments in brown highlights in pdf format. Sorry for your inconvenience with so many yellow highlights and now I removed the yellow highlights from the manuscript.

Point 1: The manuscript falls short in Section One for lack of Research Questions, which should guide the presentation of results in an orderly fashion. Please see Lines 156-158.

Response 1: Thank you for your critical comment. According to your comment, I incorporated the main research question at the end of section 1. See the additions from Lines 179 to 185 in the revised manuscript.

Point 2: Just before Section 2 (Line 159), it would be better if the author could insert a new section on Conceptual Framework. This new section should be limited to about two paragraphs only. Its purpose is to describe the conceptual underpinnings of the paper, which should also guide data analysis under the research questions.

Response 2: Thank you for your suggestion. I inserted a new section 1.2 entitled conceptual framework and elaborated the main concepts in the manuscript. For the additions see Lines 187 to 242 in the revised manuscript.

Point 3: Figure 1. comprises three smaller figures. This Figure 1 should be split as Figures 1.1, 1.2 and 1.3. under Section 2.1.

Response 3: Thanks for your suggestion. Section 2.1 describes the study area and Figure 2 showed clearly the study area map and the three maps should be put in one frame. Because the first map showed Ethiopia and the second map Amhara region is part of Ethiopia and the third one showed the East Gojjam Zone (part of Amhara region) that consisted of the selected districts in the zone (Basoliben, Shebel Berenta and Enbese Sar Midir).

Point 4: In order to enhance the analytical rigour of the paper, the whole of Section 3 (Results) must be properly restructured in tandem with the Research Questions listed in Section 1.

Response 4: Thanks for your comment and suggestion. According to your comment, I inserted a research question at the end of section 1 and based on the orderly fashion of the research questions I structured the results based on the main questions. I showed some results without the main research question because of the background results. For the improvement see lines 403 to 527 in the revised part.

Point 5: Your discussion should follow the order in the preceding Section (Results) Here, the author should compare the findings with the findings in relevant previous studies as done in Line 525.

Response 5: Thank you very much for your comments. According to your comment, I structured the discussion according to the research question and results also I compared my finding with other previous similar studies. See the improvements from lines 528 to 717 in the revised manuscript.

Point 6: The author should revisit the Conclusion, which should be a standalone section. Here, we should see a clear summary of the findings under each research question.

Response 6: Thanks for your valuable comments. I wrote the conclusion in an orderly manner on the research question and results. See lines 718 to 785 in the revised manuscript.

Point 7: The ubiquitous listing of 'Author Contributions' is very unnecessary because the manuscript has only one author.

Response 7: Thanks for your valuable comments. I deleted the Author Contributions section. See lines 789 to 792 in the revised manuscript.

Point 8: The literary composition of the paper is fairly good, but the author should correct the errors identified. 

Response 8: Thanks for your valuable comments. I corrected the identified errors. Please see the whole manuscript for the quality of the English language.

Additional Comments from the PDF document:

Line 2: certification …I changed to registration.

Line 8: What method did you use to determine the sample size? To calculate the sample size (n), the statistical formula Cochran (1977) [73] was used. With a 95% confidence level and a 5% sampling error, the sample size (n) was calculated. As a result, the study's sample size (n) was 385 households.

Line 13: 385 households is the sample size calculated by the formula but 163 households (from 385 samples) have a mean of 0.40 ha of agricultural land on a steep slope area. All 15082 total households listed in fifteen selected Kebeles were the sample frame (N).

Line 34: According to your comment I started the introduction section with land registration and for the improvement see lines 36 to 42 in the revised manuscript.

Line 50 to 54: I shortened the sentences see lines 58 to 63 in the revised part.

Line 73 I changed to by government see line 86.

Anyways I incorporated all the comments given in the pdf document.

Thank you very much for your understanding.

Reviewer 4 Report (New Reviewer)

I would like to thank the authors for a very interesting research topic, but I would like to make some corrections to improve the quality of the manuscript. The title is clear and indicates the entire research. The summary contains all the necessary elements. I would suggest that the literature review be consolidated so that the slots insert literature related to similar issues and research for each of the variables they investigated. The methodology is explained precisely. In the results section, add a discussion to the effect that the research and results are related to similar research on the given topic. Expand concluding considerations, add limiting circumstances, future theoretical and applied implications.

Author Response

Response to Reviewer 4 Comments

We would like to express our sincere thanks to the reviewer for the valuable comments/suggestions. Our point-by-point responses are given below in red colour.

Comments and Suggestions for Authors:

I would like to thank the authors for a very interesting research topic, but I would like to make some corrections to improve the quality of the manuscript. The title is clear and indicates the entire research. The summary contains all the necessary elements. I would suggest that the literature review be consolidated so that the slots insert literature related to similar issues and research for each of the variables they investigated. The methodology is explained precisely. In the results section, add a discussion to the effect that the research and results are related to similar research on the given topic. Expand concluding considerations, add limiting circumstances, and future theoretical and applied implications.

Thank you very much for your appreciation and comments. I incorporated all the comments and for details see lines 537 to 721 and Lines 731 to 799 in the revised manuscript.

Thank you very much for your understanding.

Round 2

Reviewer 3 Report (New Reviewer)

1. The most critical issue is that the research questions are not properly linked to the results. There is no proper link; hence, it is difficult to say whether or not the research questions have been answered satisfactorily.

2. The numbering of the research questions (1, 2, 3, etc.) is not acceptable. Use a, b, c, etc. instead.

3. Figure 2 is clumsy and should be clarified as advised in my comments.

4. The references need to be improved by including some more recent citations starting from at least Year 2020.

5. The conclusions and recommendations should be revised accordingly.

1. The literary composition is satisfactory. 

2. The authors are strongly advised to spend more time on the identified errors in the paper as indicated in my comments.

Author Response

We would like to express our sincere thanks to the reviewer for the valuable comments/suggestions. Our point-by-point responses are given below in red colour.

Comments and Suggestions for Authors:

Point 1: The most critical issue is that the research questions are not properly linked to the results. There is no proper link; hence, it is difficult to say whether or not the research questions have been answered satisfactorily.

Response 1: Thank you for your comment. According to your comment, I added one research question that showed the flow of the research question in line with the order of the results and discussions. Look research question number 1 was answered by the result section put in section 3.3; question 2 by result section 3.4; question 3 was answered by result section 3.6; question 4 was answered by result section 3.7.1; and last research question 5 was answered in the result section 3.7.2. See the additions from Lines 173 to 174  and the clarification of lines 434 to 504 in the revised manuscript.

Point 2: The numbering of the research questions (1, 2, 3, etc.) is not acceptable. Use a, b, c, etc. instead.

Response 2: Thank you for your suggestion. I changed to a, b, c, d and e. And also I added one research question. For the corrections see Lines 166 to 177 in the revised manuscript.

Point 3: Figure 2 is clumsy and should be clarified as advised in my comments.

Response 3: Thank you for your suggestion. In response to your comment, I've changed Figure 2 so that the three maps are all in one border frame, and I've drawn a path for the maps' flow. For example, from Ethiopian regions, the study area was found in Amhara, so the arrow was placed in the second map that showed Amhara regions zones, and the arrow was placed in Misrak (East) Gojjam, then the third map is the exact study site that is East Gojjam zone, and the arrow showed the exact area. Also from East Gojjam, I selected three districts in different colours: Enbese Sar Midir, Shebel Berenta, and Basoliben. Line 268 of the revised manuscript contains the corrections.

Point 4: The references need to be improved by including some more recent citations starting from at least the Year 2020.

Response 4: Thanks for your comment and suggestion. I tried to cite recent citations.

Point 5: The conclusions and recommendations should be revised accordingly.

Response 5: Thank you very much for your comments. I tried to revise the conclusion section.

Thank you very much for your understanding.

Round 3

Reviewer 3 Report (New Reviewer)

1, A few research questions have been poorly constructed, e. g. Starting with "Did...." instead of "Does....." The editors need to pay more attention to this section of the paper.

2, Some words have been separated by hyphen unnecessarily. 

3. The authors still need to try and see if they can add about 3 or 4 more recent citations (after Year 2020). We are in Year 2023 today. I think that a top research paper should be very current.

4. I am not comfortable with the authors' 

Author Response

We would like to express our sincere thanks to the editor and reviewer for the valuable comments/suggestions. Our point-by-point responses are given below in red colour.

Comments and Suggestions for Authors:

Point 1: A few research questions have been poorly constructed, e. g. Starting with "Did...." instead of "Does....." The editors need to pay more attention to this section of the paper.

Response 1: Thank you for your comment. I corrected the research questions. For the correction see lines 170 to 178.

Point 2: Some words have been separated by hyphen unnecessarily.

Response 2: Thanks. Accordingly, I removed the unnecessary hyphen.

Point 3: The authors still need to try and see if they can add about 3 or 4 more recent citations (after Year 2020). We are in Year 2023 today. I think that a top research paper should be very current.

Response 3: Thank you for your suggestion. I added more recent citations. For example: Likewise, land registration has strong implications for household participation in sustainable land management initiatives at the community level [84]. Most of the land tenure regularization programs predict an increase in land-based investments such as soil and land management infrastructure due to land registration and certification [85]. Land management incentives promote the positive impact of the land registration program [86]. Additionally, studies done in Damot-Gale District, Southern Ethiopia revealed that the majority (62%) of the respondents indicated that they are practising land management due to a certificate, i.e., land certificate increases the perception of farmers in Land Management Practices [87]. So, see reference numbers 84 to 87 in lines 967 to 974.

Point 4: I am not comfortable with the authors'.

Response 4: Dear Reviewer, To the best of my knowledge, I attempted to incorporate your comments and suggestions completely. Please see the revised manuscript for the incorporations.

MDPI comments:

Point 1: Line 17: … of households and? Respectively. Where is the second variable used for comparison here?

Response 1: Thank you for your important comment and I rewrite it as approximately 26% of households are afraid of land redistribution and farm loss in the next five years. Moreover, 22% of households fear the government taking their farm plot at any time. See lines 17 to 19.

Point 2: Line 35: Don’t leave a gap before this first line. You can do so for other cases.

Response 2: Thanks. I edited for the gap issue before the first line. See the revised manuscript for the edition.

Other important comments are shown in the pdf.

Line 22: mean comparison…

I corrected the difference in mean of…

Line 174: Conceptual Framework:

I corrected this part.

Line 345: The equation of the logit model should rather appear under model specification, not above it.

Thanks, but the equation must be put after the data analysis technique and it is the right place for the equation. The model specification only talks about the description of the dependent variables.

Line 382: Results: We need to see how you utilised each of your research questions. Start with each RQ and put your findings down serially. Do so in an orderly version. This makes your arguments logical. See your list of RQ above.

Thanks. Now according to the order of the research questions I put the results down serially and all research questions were answered successfully.

Line 477: Results of the logit model. Which of the RQs is this section 3.7 addressing

Thanks. For research questions 4 and 5.

Line 504: Again, when discussing your results, always remember to do so with recourse to a specific RQ. Let there be ORDER!

Thank you. The flow or order of the discussion is also according to the order of the research questions and results. For clarification see the revised manuscript.

Additional responses to round two comments for reviewer three:

Point 1: The most critical issue is that the research questions are not properly linked to the results. There is no proper link; hence, it is difficult to say whether or not the research questions have been answered satisfactorily.

Response 1: Thank you for your comment. The flow of research questions is in line with the order of the results and discussions. Look research question number 1 was answered by the result section put in section 3.3; question 2 by result section 3.4; question 3 was answered by result section 3.6; question 4 was answered by result section 3.7.1; and last research question 5 was answered in the result section 3.7.2. See the research questions from lines 170 to 178, the results from lines 389 to 507 and discussions from lines 510 to 707 in the revised manuscript.

Point 2: The numbering of the research questions (1, 2, 3, etc.) is not acceptable. Use a, b, c, etc. instead.

Response 2: Thank you for your suggestion. I changed to a, b, c, d and e.

Point 3: Figure 2 is clumsy and should be clarified as advised in my comments.

Response 3: Thank you for your suggestion. Now, in response to your comment, I've changed Figure 2 so that the three maps are all in one border frame, and I've drawn a path for the maps' flow. For example, from Ethiopian regions, the study area was found in Amhara, so the arrow was placed in the second map that showed Amhara regions zones, and the arrow was placed Misrak (East) Gojjam, then the third map is the exact study site that is East Gojjam zone, and the arrow showed the exact area. Also from East Gojjam, I selected three districts in different colours: Enbese Sar Midir, Shebel Berenta, and Basoliben. Line 268 of the revised manuscript contains the corrections.

Point 4: The references need to be improved by including some more recent citations starting from at least Year 2020.

Response 4: Thanks for your comment and suggestion. See the response to round three above.

Point 5: The conclusions and recommendations should be revised accordingly.

Response 5: Thank you very much for your comments. I revised the conclusion and recommendation section. For example, The findings also provide important policy implications and suggest that policymakers both at governmental and non-governmental agencies engaged in sustainable land management among rural agricultural households that aim to boost agricultural development should consider land registration as an important prerequisite. The evidence shows that if farm households are given more secure property rights on their land, they would be encouraged to increase their investments in sustainable land management. Thus, policymakers in Ethiopia should consider land registration as a matter of priority to ensure the success of sustainable land management programs and to promote the development of modern agriculture. Tenure security by ensuring the probability of benefiting from their investment in the long- term. Thus, tenure security can also serve as an incentive mechanism for the success of sustainable land management. For this see lines 768 to 778.

This manuscript is a resubmission of an earlier submission. The following is a list of the peer review reports and author responses from that submission.

Round 1

Reviewer 1 Report

Though it is a repetitive work but still not done properly. The language of the paper is not up to the mark. However, I have some specific comments :

(1) why do you take 300 sample size. The rule of thumb says if the population is above 10,000, then the sample size must be 384 with 95% confidence interval.

(2) In the methodology section, it is logit model but in the results and discussion section, it is probit model. This shows how serious are the authors.

(3) The results are not discussed adequately to bring out the novelty of the work.

(4)The whole presentation in the paper must improve to get it published in international journal.

(5) Many explanatory variable may co-evolve. So there might be endogeneity problem which should be taken care of.

(5) Conclusion must be improved.

Author Response

Response to Reviewer 1 Comments

We would like to express our sincere thanks to the reviewer for the valuable comments/suggestions. Our point-by-point responses are given below in red colour.

Comments and Suggestions for Authors:

Though it is repetitive work but still not done properly. The language of the paper is not up to the mark.

Thank you for your comment. According to your suggestion, I corrected the writing in good English that is readable. See the revisions of language and novelty starting from line 43 to the end of the manuscript. I hope the revisions would be sufficient to meet with your approval.

However, I have some specific comments :

Point 1: why do you take 300 sample sizes? The rule of thumb says if the population is above 10,000, then the sample size must be 384 with a 95% confidence interval.

Response 1: Thank you for your comment. There are various sample size calculation formulas. The Cochran sample size formula, for example, will get 343 respondents. Despite the fact that I took 300 household respondents, the method of getting 300 is appropriate for the statistics. I obtained the 300 samples through stratified random sampling from a total population of 15082. Furthermore, the number of households selected for interview from each kebele was determined proportionally to the total number of households in the kebele (rounded to an integer) to guarantee an equal representation of farm households in each kebele. Therefore, each household in her/his strata had an equal chance of being selected. For clarification see Table 1 in line 264 (revised manuscript). I hope the revisions would be sufficient to meet with your approval.

Point 2: In the methodology section, it is the logit model but in the results and discussion section, it is the probit model. This shows how serious are the authors.

Response 2: Thank you for your suggestion. I corrected the sub-title of the result section line 474 Results of the logit model.

Point 3: The results are not discussed adequately to bring out the novelty of the work.

Response 3: Thanks for your valuable comments. I discussed the results adequately according to your comment. See the revisions starting from line 387 in the revised manuscript.

Point 4: The whole presentation in the paper must improve to get it published in an international journal.

Response 4: Thanks for your comment and suggestion. I perfectly revised the whole manuscript according to your valuable comment. See the improvements in the revised manuscript starting from Line 8. I hope the revisions would be sufficient to meet with your approval.

Point 5: Many explanatory variables may co-evolve. So there might be an endogeneity problem which should be taken care of.

Response 5: Thank you very much for your concern. But the model is linear regression and also the two models are separate. That is perceived tenure security in the first model as the dependent variable and construction of the water harvesting system in the second model as the dependent variable. Also, I checked Multi-collinearity and Variance Inflation Factor (VIF) for the models. So, I assured you that there is no endogeneity problem.

Point 6: The conclusion must be improved.

Response 6: Thanks for your valuable comments. I totally revised the conclusion according to your comment. See the improvements in the revised manuscript starting from Line 723. I hope the revisions would be sufficient to meet with your approval.

Thank you very much for your understanding.

Reviewer 2 Report

In this article, the author explores the impacts of the Rural Land Certification process in the East Gojjam Zone, Ethiopia. The author uses a mix of quantitative and qualitative methods to investigate the results of a field-based survey of agriculturalists in the study regions to assess if there is a noticeable shift in land-use practice variance between certified and un-certified land occupancy. This article provides an interesting case study for this type of issue and the analytical approach to the topic is sound.  in general, seems to be a quantification of what one would expect--if a farmer's land holding was certified, they would take on more responsibility and environmental stewardship practices to increase productivity. While the results of this article is not inherently ground breaking, it does add a layer of logical support to the benefits for a shift in policy toward agricultural land certification across the marginalized communities in Ethiopia. One question that does come to mind, however, is what is the political will for/against this process in the government?

In general, the literature used throughout this article is extensive. However, the introduction section as two apparent issues. First, there is an aesthetic of over-citation whereby nearly every sentence has multiple citations. Second, even though there is a wealth of sources used, the introduction needs more direct source usage and engagement. More needs to be done to expand and explain what these sources say and how this article builds off or challenges these sources. It reads too general in the literature review and looks to have gratuitous citations trailing at the end of each main idea. Citations are fine, but their relevance and significance is overly ambiguous.

A few major issues related to the context of land redistribution and certification, in my opinion, need to be explained better throughout the article. Throughout the entire article I found myself asking “are farmers cultivating non-certified land considered illegal occupants (making it easier for the government to redistribute land), or are they legal owners of the land and certification here is more of an overlay further recognizing ownership and/or practices (sustainable)?” Line 37 brings in the idea of land certification, without actually discussing what the term means, and this also holds true wit land tenure and titling in lines (lines 49 and 55). What is missing is a needed description of percentages of land ownership rates today (public/private/corporate) as well as a discussion of if this certification/redistribution is an issue for today's agriculture sector or a long-term, historical issue? In this case what is the historical/colonial period land ownership dynamic in Ethiopia and how it relates to today would be helpful for context. Was any of this redistribution/certification influenced by a long response to the 1970s/1980s famines or something else? Further discussion related to the shift in 1998 toward certification is needed too. Why was there a policy shift? What was the political justification and context? Another detail that is needed is examples of regional or national redistribution that have happened in past/recently and #of acres/hectares? Similarly, how many hectares/acres have been certified since 1998? Another question I had throughout was description of the process of how the government actually conducts a land redistribution is needed. It is a bit abstractly discussed in this article, but is a pressing fear for farmers. Is it random, targeted (infrastructure development), selling to foreign investor like Saudi Arabia? Also, how has the recent Tigray conflict influenced this system of land redistribution and certification if the various regions? Finally, is any of this push toward certification being done in accordance with international initiatives like the UN’s SDGs?

As far as the discussion of the data collection methods and statistical analysis methods used, they are good. The discussion of each table and the responses is good. However, I feel as though a few aspects are missing. As far as 20 households per kebele (a term that needs to be further defined early on in the article), why was this number chosen? How many households are there in total in the study area? What are the response rates and confidence rates based on the 20 households? In section 2.2.2, the HHS survey (I assume conducted by the Ethiopian government) needs to be discussed more, especially what data the source provides. In addition, I would have liked to see more of the qualitative responses from the focus groups to personalize the quantitative data/tables. The voices of those involved need to be more clearly accounted for rather than just the numeric responses. In the final discussion section, I think the direction of the discussion is sound, but it could also benefit from placing it further into the context of sustainability, and even taking the discussion f the impacts of the findings to a larger international context (UN SLOs or UNCCD scale). However, keeping the discussion related to the specifics of Ethiopia is fine as well.

Author Response

Response to Reviewer 2 Comments

Thank you, dear reviewer, for your valuable comments/suggestions to enrich the paper. We tried to address all issues to improve the paper. Replies to each review comment are given below in red colour.

Comments and Suggestions for Authors:

Point 1: In this article, the author explores the impacts of the Rural Land Certification process in the East Gojjam Zone, Ethiopia. The author uses a mix of quantitative and qualitative methods to investigate the results of a field-based survey of agriculturalists in the study regions to assess if there is a noticeable shift in land-use practice variance between certified and un-certified land occupancy. This article provides an interesting case study for this type of issue and the analytical approach to the topic is sound. in general, seems to be a quantification of what one would expect--if a farmer's land holding was certified, they would take on more responsibility and environmental stewardship practices to increase productivity. While the results of this article are not inherently groundbreaking, it does add a layer of logical support to the benefits of a shift in policy toward agricultural land certification across the marginalized communities in Ethiopia. One question that does come to mind, however, is what is the political will for/against this process in the government?

Response 1: Thank you very much for your affirmation and many detailed and valuable comments and question. “the results of this article is not inherently groundbreaking, it does add a layer of logical support to the benefits of a shift in policy toward agricultural land certification across the marginalized communities in Ethiopia”. For results indicating the policy of marginalised societies see lines 432 to line 450 and lines 531 to line 555 in the revised manuscript.

What is the political will for/against this process in the government? The government of Ethiopia favours marginalized communities in the certification process. For instance, the revised Amhara Region Land Administration and Use (ARLAU) Proclamation No. 133/2006 Article 9 (2) supports land holding in priority order for orphans, the disabled, women, and young people who join the new life of independence. For the detail see lines 541 to line 552 in the revised manuscript.

Point 2: In general, the literature used throughout this article is extensive. However, the introduction section has two apparent issues. First, there is an aesthetic of over-citation whereby nearly every sentence has multiple citations. Second, even though there is a wealth of sources used, the introduction needs more direct source usage and engagement. More needs to be done to expand and explain what these sources say and how this article builds off or challenges these sources. It reads too general in the literature review and looks to have gratuitous citations trailing at the end of each main idea. Citations are fine, but their relevance and significance are overly ambiguous.

Response 2: I thank the reviewer for the valuable suggestions. So, I confirmed that there is no over-citation and I used the first (direct) source. All citations used are significant and supportive of the study. For the details see lines 43 to line 194 in the revised manuscript.

Point 3: A few major issues related to the context of land redistribution and certification, in my opinion, need to be explained better throughout the article. Throughout the entire article, I found myself asking “are farmers cultivating non-certified land considered illegal occupants (making it easier for the government to redistribute land), or are they legal owners of the land and certification here is more of an overlay further recognizing ownership and/or practices (sustainable)?” Line 37 brings in the idea of land certification, without actually discussing what the term means, and this also holds true with land tenure and titling in lines (lines 49 and 55). What is missing is a needed description of percentages of land ownership rates today (public/private/corporate) as well as a discussion of if this certification/redistribution is an issue for today's agriculture sector or a long-term, historical issue. In this case, what is the historical/colonial period land ownership dynamic in Ethiopia and how it relates to today would be helpful for context. Was any of this redistribution/certification influenced by a long response to the 1970s/1980s famines or something else? Further discussion related to the shift in 1998 toward certification is needed too. Why was there a policy shift? What was the political justification and context? Another detail that is needed is examples of regional or national redistribution that have happened in past/recently and #of acres/hectares. Similarly, how many hectares/acres have been certified since 1998? Another question I had throughout was a description of the process of how the government actually conducts land redistribution is needed. It is a bit abstractly discussed in this article but is a pressing fear for farmers. Is it random, targeted (infrastructure development), selling to foreign investors like Saudi Arabia? Also, how has the recent Tigray conflict influenced this system of land redistribution and certification of the various regions? Finally, is any of this push toward certification being done in accordance with international initiatives like the UN’s SDGs?

Response 3: Thanks for your question, suggestion, and valuable comments. “Are farmers cultivating non-certified land considered illegal occupants (making it easier for the government to redistribute land), or are they legal owners of the land and certification here is more of an overlay further recognizing ownership and/or practices (sustainable)?” Again, thank you for your question. In Ethiopia, second-level Land certification is still in the process and after some time all farms will be certified. Regarding my study, I took cases that are uncertified but after some time will be certified and get a land certificate. Up to now in Ethiopia as a whole 15 million parcels of the total 50 million parcels had been registered and certificates distributed to landholders. For the details see lines 148 to line 153 in the revised manuscript.

Line 37 brings in the idea of land certification, without actually discussing what the term means, and this also holds true with land tenure and titling in lines (lines 49 and 55). What is missing is a needed description of percentages of land ownership rates today (public/private/corporate) as well as a discussion of if this certification/redistribution is an issue for today's agriculture sector or a long-term, historical issue. In this case, what is the historical/colonial period land ownership dynamic in Ethiopia and how it relates to today would be helpful for context. For the revisions see lines 45 to line 50 and lines 69 to line 73. For the certified parcels list see lines 148 to line 153. Regarding the historical issues of the land tenure system in Ethiopia see lines 101 to line 115 in the revised manuscript.

Was any of this redistribution/certification influenced by a long response to the 1970s/1980s famines or something else? Further discussion related to the shift in 1998 toward certification is needed too. Why was there a policy shift? What was the political justification and context? For the history of redistribution see lines 120 to 135 in the revised manuscript. Recognizing the negative effects of tenure insecurity on long-term land management practices, the Ethiopian government launched a large-scale land registration and certification program in 1998. Ethiopia has one of Africa's largest, fastest, and least expensive land registration and certification reforms. Which has been credited as being a benchmark for land certification in Africa. Additionally, the government positively favoured land certification and also put it in the policy document.

Another detail that is needed is examples of regional or national redistribution that have happened in past/recently and #of acres/hectares. Similarly, how many hectares/acres have been certified since 1998? Another question I had throughout was a description of the process of how the government actually conducts land redistribution is needed. It is a bit abstractly discussed in this article but is a pressing fear for farmers. Is it random, targeted (infrastructure development), selling to foreign investors like Saudi Arabia? Also, how has the recent Tigray conflict influenced this system of land redistribution and certification of the various regions? Finally, is any of this push toward certification being done in accordance with international initiatives like the UN’s SDGs?

See lines 132 to 138 for information on land redistribution. Because many landholder farmers have more than 5 hectares of land and others, such as youth, women, and others, do not, the government conducted land redistribution in 1997. As a result, at the time, the government redistributed land and gave it to landless youth, orphans, and women. As a result, the goal is not to give to investors or to develop infrastructure. Another point you raised was that the recent Tigray conflict had no influence on land redistribution. For the UN SDGs goal of certification, see lines 63 to line 70 in the revised manuscript. Anyways, I revised the manuscript according to your detailed suggestions and comments. I hope these revisions would be sufficient to meet with your approval.

Point 4: As far as the discussion of the data collection methods and statistical analysis methods used, they are good. The discussion of each table and the responses is good. However, I feel as though a few aspects are missing. As far as 20 households per kebele (a term that needs to be further defined early on in the article), why was this number chosen? How many households are there in total in the study area? What are the response rates and confidence rates based on the 20 households? In section 2.2.2, the HHS survey (I assume conducted by the Ethiopian government) needs to be discussed more, especially what data the source provides. In addition, I would have liked to see more of the qualitative responses from the focus groups to personalize the quantitative data/tables. The voices of those involved need to be more clearly accounted for rather than just the numeric responses. In the final discussion section, I think the direction of the discussion is sound, but it could also benefit from placing it further into the context of sustainability, and even taking the discussion f the impacts of the findings to a larger international context (UN SLOs or UNCCD scale). However, keeping the discussion related to the specifics of Ethiopia is fine as well.

Response 4: Thank you very much for your affirmation and many detailed and valuable comments including questions. As far as 20 households per kebele (a term that needs to be further defined early on in the article), why was this number chosen? How many households are there in total in the study area? What are the response rates and confidence rates based on the 20 households? Out of the 15082 households listed in fifteen selected kebeles (see the detail in Table 1 line 264 in the revised manuscript), 300 households (of whom 172 joint landholders, 53 male landholders and 75 female landholders) were randomly sampled for face-to-face interviews using a structured questionnaire. The number of households selected for interview from each kebele was determined proportionally to the total number of households in the kebele (rounded to an integer) to guarantee an equal representation of farm households in each kebele. Therefore, each household in her/his strata had an equal chance of being selected. For detail see lines 226 to line 266 in the revised manuscript.

In section 2.2.2, the HHS survey (I assume conducted by the Ethiopian government) needs to be discussed more, especially what data the source provides. Household Surveys (HHS) conducted from September/2022 to October/2022 were the primary source of data. Primary and secondary data were collected in this study. For the field interviews, both closed and open-ended structured questionnaires were used to collect primary data. Structured questionnaires were developed, tested, and adjusted to fit their intended purpose. Farmers were asked about their perceptions of land-holding rights and land management activities before and after certification. For detail on data collection see lines 267 to line 301 in the revised manuscript.

I would have liked to see more of the qualitative responses from the focus groups to personalize the quantitative data/tables. The voices of those involved need to be more clearly accounted for rather than just the numeric responses. I added qualitative responses and sustainability concepts in the discussion section. for the detail of the discussion see lines 514 to line 721 in the revised manuscript. Generally, I revised the whole manuscript according to your detailed suggestions and comments. Hopefully, these revisions would be sufficient to meet with your approval.

Thank you very much for your understanding.

Round 2

Reviewer 2 Report

I think the changes made to this article are good. It better clarifies the land tenure system in Ethiopia as well as frames the content in the larger discussion of the UN's SDGs. It also better situates the land tenure policies within the context of the Ethiopian government agencies active in this issue. I like the reconfiguration and expansion of the introduction as well as the changes to the conclusion. There are a few incomplete sentences in the revised sections of the text as well as some spacing issues every so often. All and all, these changes make the article much easier to understand both contextually and methodologically. 

Author Response

Thank you, dear reviewer, for your valuable comments/suggestions to enrich the paper. We tried to address all issues to improve the paper. Replies to each review comment are given below in red colour.

Comments and Suggestions for Authors:

Point 1: I think the changes made to this article are good. It better clarifies the land tenure system in Ethiopia as well as frames the content in the larger discussion of the UN's SDGs. It also better situates the land tenure policies within the context of the Ethiopian government agencies active in this issue. I like the reconfiguration and expansion of the introduction as well as the changes to the conclusion. There are a few incomplete sentences in the revised sections of the text as well as some spacing issues every so often. All and all, these changes make the article much easier to understand both contextually and methodologically.

Response 1: Thank you very much for your appreciation and comments. The manuscript has been revised in response to your feedback. I rewrite the incomplete sentences and correct the spacing issues. See the revised manuscript for more information.

Thank you very much for your understanding.
